# Strategies to Achieve Breast Health Equity in the St. Louis Region and Beyond over 15+ Years

**DOI:** 10.3390/cancers14102550

**Published:** 2022-05-23

**Authors:** Bettina Drake, Aimee James, Heidi Miller, Akila Anandarajah, Kia L. Davis, Sheryll Jackson, Graham A Colditz, Vetta Sanders Thompson

**Affiliations:** 1Division of Public Health Sciences, Department of Surgery, Washington University School of Medicine, St. Louis, MO 63110, USA; drakeb@wustl.edu (B.D.); aimeejames@wustl.edu (A.J.); akila@wustl.edu (A.A.); daviskl@wustl.edu (K.D.); sherrilljackson23@gmail.com (S.J.); 2Alvin J. Siteman Cancer Center, Barnes-Jewish Hospital and Washington University School of Medicine, St. Louis, MO 63110, USA; vthompson22@wustl.edu; 3St. Louis Integrated Health Network, St. Louis, MO 63118, USA; millerheidib@gmail.com; 4Brown School, Washington University in St. Louis, St. Louis, MO 63130, USA

**Keywords:** breast cancer, community-based participatory research, health disparities, mobile mammography

## Abstract

**Simple Summary:**

Breast cancer is a leading cause of death in women in the United States. However, there exist inequalities in outcomes of breast cancer and medically underserved populations by race, insurance status, educational background, and geography. Here, we present efforts by the Siteman Cancer Center to address disparities over the past 15+ years. As an academic center, we have partnered with community organizations to help support patients and reduce barriers to receiving care. We also ran a mobile van to offer mammograms to women living in both urban and rural areas of Missouri. These efforts resulted in earlier detection of breast cancer and increased use of mammograms. The mammography van managed to reach mostly poor, uninsured, or underinsured women with limited educational backgrounds. This work demonstrates the potential for collaborations between academic and community partners to improve community health outcomes.

**Abstract:**

Community-based participatory strategies are a promising approach to addressing disparities in community health outcomes. This paper details the efforts of Siteman Cancer Center to achieve breast health equity over the past 15+ years. We begin by describing the activities and successes arising from our breast health community partnerships including identifying priorities, developing recommendations, and implementing patient navigation services to advance breast health. This system-wide coordinated navigation approach that includes primary and specialty care providers helped to increase potential impact on reducing breast health disparities by expediting care, increasing care efficiency, and standardizing referral procedures across systems for all women including those who are uninsured and underinsured. We also discuss a mobile mammography unit that has been deployed to serve women living in both urban and rural regions. The van reached a particularly vulnerable population that was mostly poor, uninsured, and with limited educational backgrounds regardless of their zip code of service. This work shows that collaborations between academic and community partners have resulted in decreased late stage at diagnosis and improved access to mammography. Furthermore, we offer lessons learned and recommendations that may be applicable to other communities.

## 1. Introduction

Breast cancer continues to be the most commonly diagnosed cancer in the United States, and it is the second leading cause of death among women in the United States. The National Cancer Institute’s Surveillance, Epidemiology, and End Results Program (SEER) cancer statistics report estimates that over 280,000 women will be diagnosed with breast cancer and nearly 45,000 will die of complications related to the disease in 2021 alone [1]. A proven method to reduce breast cancer mortality is mammography, with a 15–30% reduction in breast cancer specific mortality, depending on the age of the patient [2]. This reduction in mortality is based on mammography’s ability to detect clinically asymptomatic breast cancers at an early stage, leading to early diagnosis and initiation of treatment. Despite the proven effectiveness of mammography, the most recent data from National Health Interview Survey reports demonstrated that only 70.7% of women between the ages of 45–64 had completed a mammogram within two years, which was significantly below the Healthy People 2030 goal of 77.1% [3,4]. In this report, factors associated with not having undergone a mammogram included lower educational attainment, lower income, and not having insurance.

However, there are disparities in breast cancer incidence and mortality. Black, Hispanic, and American Indian or Alaskan Native women are more likely to be diagnosed at a more advanced stage than White women [5]. While Blacks have lower average breast cancer incidence rates than Whites, they have much higher average mortality rates nationwide [6]. They also have higher incidence of triple negative breast cancer, an aggressive subtype that is more prone to recurrence and is less responsive to treatment than it is for other race/ethnic groups [7], with lower survival after treatment [8]. These differences are exacerbated in the St. Louis, Missouri region where the Siteman Cancer Center is located. 27% of patients live in a medically underserved area and 13% live in a rural area [9]. Breast cancer incidence and mortality in both St. Louis County and Missouri are above the national average. In St. Louis City, Black women are more likely to have had at least one mammogram ever and in both Missouri and Illinois, they are more likely to have had a mammogram within the past two years than White women [10,11]. Despite this, they are almost twice as likely to die from breast cancer. Factors that could explain these health inequities include disparities in healthcare access and quality, lack of quality insurance, higher late-stage diagnoses, and the higher prevalence of triple negative breast cancer [7,11]. This paper details the community-based participatory strategies employed by Siteman Cancer Center to reduce breast cancer disparities over the past 15+ years including the use of community partnerships and a mobile mammography unit.

## 2. Breast Health Community Partnership

In response to concerns about disparities in cancer outcomes in our region, the Siteman Cancer Center launched the Program for the Elimination of Cancer Disparities (PECaD) in 2003. PECaD works through community partnerships to develop outreach and education, quality improvement and research, and training strategies to lessen the burden from cancer disparities. PECaD’s target geographic area for addressing breast cancer disparities is the St. Louis metro area. From 2003–2007, PECaD’s infrastructure was established and ideas for community advisory groups arose, with PECaD also working on other cancer sites including the colon and prostate. Additionally, a needs assessment was conducted in each of these areas. Ignited by alarmingly high breast cancer mortality rates in the St. Louis region, particularly in north St. Louis City and County among Black women, PECaD’s community advisory board advocated for research which showed high rates of late-stage breast cancer diagnosis, which could help explain the mortality disparity [12]. The Breast Cancer Community Partnership was established in 2007 with the objective of fostering ongoing dialogue about breast cancer disparities between community stakeholders and academic faculty, setting priorities and developing solution strategies concerning local and regional breast cancer disparity issues. Members of the partnership include lay community leaders, breast cancer survivors, advocates, health care organizations, community-based organizations, community support groups and academic faculty and staff.

In 2007, building on previous work from members over the past seven years, the Breast Cancer Community Partnership identified seven priorities for the region to advance breast health among uninsured and underinsured women. They were to: (1) Create greater awareness related to prevention and access to free or low-cost care among lay community members as well as providers; (2) Understand the role/impact of culture on breast health behaviors; (3) Build strong relationships and trust between the community and academics; (4) Identify community and clinical strategies to help patients keep breast health appointments; (5) Increase adherence to routine breast cancer screening; (6) Examine and disseminate best practices in health literacy and health communication to support breast health; and (7) Evaluate regional impact. For the remainder of this paper, we will focus primarily on programs and activities addressing these priorities (Table 1).

In order to address the priorities to identify strategies to help patients keep breast health appointments and increase adherence to screening, members of the Breast Cancer Community Partnership recommended a study to examine referral processes and to identify individual behaviors, structural factors, and institutional practices and policies that might contribute to delays in breast cancer diagnosis and treatment and stage at diagnosis. As a joint effort between community and academic institutions, we led an evaluation of screening and diagnostic efficiency across the entire safety-net system that provides services to uninsured and underinsured women including primary and specialty care provision. Findings using data from between 2005 and 2007 showed a need to expedite referral processes for all women within the system and to develop strategies addressing patient-specific barriers to breast care adherence [13]. In direct response to these findings, a close community partner, the St. Louis Integrated Health Network, led the Breast Cancer Referral Initiative to identify opportunities for improving referral processes for patients in need of breast cancer treatment [14]. Members included primary and specialty care representatives of the safety-net system, including community and academic provider organizations, as well as key individuals from PECaD’s Breast Cancer Community Partnership. The workgroup met from January to October 2009 and developed recommendations to expedite all processes of breast health screening and diagnosis of breast cancer within the safety network. These recommendations were designed to minimize health care disparities among women in St. Louis diagnosed with breast cancer.

Patient navigators are healthcare staff or community volunteers who support patients through all components of care while reducing barriers related to insurance, access, mistrust, fear, culture, and social determinants of health. Patient navigation interventions have become a widely used and effective approach to advance and extend services in public health and health care, especially for chronic disease management and with vulnerable populations. They are effective for increasing cancer screening rates, with the majority of interventions focused on vulnerable populations [15]. The parallel efforts of the breast cancer evaluation findings, the Breast Cancer Community Partnership discussions, and the recommendations of the Breast Cancer Referral Initiative converged on the consensus that (1) breast health navigation is critical to breast health equity, and (2) it is essential for breast health navigators from disconnected organizations to collaborate on behalf of the same patient population. This was deemed essential to advance breast health quality improvements in a manner that spanned all stages of the breast cancer care continuum, united providers from primary and specialty care, and connected community and academic institutions. As a result, PECaD established the St. Louis Regional Breast Navigator Workgroup (RBNW) in 2010 with the charge to:Improve communication between regional safety-net providers; andDevelop more effective and efficient processes for breast cancer screening, referral, diagnosis, treatment, and follow-up/survivorship in the region.

The workgroup consists of key breast health navigators and advocates involved with breast health referrals across the cancer continuum from screening through survivorship. The navigators represent over 30 organizations that outreach to the St. Louis MO-IL safety-net system, including community-based healthcare providers, safety-net organizations, diverse hospital systems, regional breast health centers, universities, breast health nonprofits, and other breast health advocacy organizations. Consistent with the patient-navigation literature [15], navigators of the RBNW have heterogeneous titles within their respective organizations including navigator, case manager, nurse, social worker, community health worker, medical assistant, referral coordinator and others. PECaD has established, supported, and advanced this successful collaborative workgroup since its inception, providing administrative and logistical support, securing facilitator time, and fostering movement and progress on discussion topics toward the group’s charge. In our efforts to improve the care pathways to breast cancer screening and timely diagnosis for women in the St. Louis safety net, we have continued to underwrite the quarterly convening of regional breast health navigators to share best practices, communicate challenges, and discuss solutions to improve the system of care for breast cancer screening and diagnosis.

The RBNW identified system-wide areas for improving opportunities to expedite breast cancer referrals for safety-net patients. These had several benefits to underserved women, including a reduced delay in diagnosis, efficient and expedited care, standardized referral procedures for all women, and system-wide coordination across primary and specialty care and inside and outside of a single academic institution. This population approach maximizes advantages across the entire system and has a larger potential for reducing breast cancer disparities.

Our targeted efforts included partnering with the Betty Jean Kerr People’s Health Centers, a local Federally Qualified Health Center (FQHC), and a partner of the Breast Cancer Community Partnership to increase adherence to routine breast cancer screening and evaluate regional impact. 87% of the patient population of this FQHC is African American. Given that 88% of eligible women at the FQHC had not received a mammogram in 2008, we received American Recovery and Reinvestment Act funding to implement and evaluate breast navigation among women due or overdue for mammography from 2009–2011 [16]. Using these funds allocated for creating jobs and sustainability, we hired a patient navigator, mammography technologist, and data coordination assistant at this underserved location. Patient navigators connected women with mammography, diagnostic, and treatment services, guided women through all aspects of care including follow-up visits and contacted women who had lost contact with the clinic. Patient navigation promoted mammogram utilization with an increased number of mammograms provided overall. These efforts resulted in 94.8% of all eligible women receiving navigation. Of the women navigated, 94.5% received a mammogram. In addition, after one and two years of implementing mammography and navigation services, 17.7% and 27.6% of women over 40 years old received a mammogram, respectively, which was an increase from 12% in the nine-month period before the study started. Navigation services continue to be offered alongside clinical care as an ongoing process with the created roles now funded by the FQHC. In the following six months after the study was conducted, 18.7% of women aged 40 or older at the FQHC received a mammogram. Importantly, these navigation and mammography services have been sustained and have expanded to navigation for additional services including annual well visits and pediatric appointments.

Throughout the region, we have also made efforts to bring awareness to prevention and free or low-cost services such as patient navigation and mammography as well as strengthen relationships between the community and academics. Awareness was initially spread through health fairs, newspaper, billboards, and public libraries, and later also through the Community Research Fellows Training program. The Community Research Fellows Training (CRFT) program is an evidence-based, 15-week capacity building research training program which uses a public health approach to equip community members to better engage with research. CRFT seeks to increase community understanding of how to use research to improve community health outcomes and increase community collaborations with academic researchers. As such, the CRFT program serves as a connector, providing the infrastructure to develop new partnerships, and a catalyst, providing the foundation for the development of new and innovative projects to address health disparities [17,18]. Evaluations of the CRFT program provide evidence that it increases participants’ knowledge of public health research between baseline and follow-up [17,19,20,21].

Since 2013, the program has graduated more than 170 community members from all areas of St. Louis and the Siteman Cancer Center catchment areas in Illinois and rural Missouri. CRFT cohort members are now integrated into the PECaD infrastructure. Members have joined the Breast Cancer Advisory Partnership (BCaP) as well as the Colorectal Cancer Partnership. CRFT participants in BCaP represent two Black women-led organizations providing breast cancer education and resources. In addition, CRFT members have leadership roles in PECaD, serving on the Disparities Elimination Advisory Council and as the community co-chair of BCaP.

Through PECaD, we have engaged FQHCs, revised and implemented screening and referral complemented by mammography outreach, and increased community awareness of breast health services. This has resulted in a decrease in stage of diagnosis, improved access to mammography, and a sustained navigator network. Diagnoses of breast cancer among Black women were 32% stage III/IV in 2000. By 2019, expanded services and partnerships reduced late-stage diagnosis to 16.4% of incident cancers among Black women, still higher than the 9.8% observed among White women.

However, there is still work to be done to enhance the navigation process across the cancer continuum through survivorship. Our previous work showed that fragmented navigation with separate or no navigators in different departments hindered completing the prescribed treatment [22]. More cohesive navigation could help reduce disparities and increase adherence to treatment.

## 3. Mobile Mammography Van

Part of the challenge in achieving breast health equity is bringing mammography closer to the women who need it. Missouri officially has 114 counties and one independent city (St. Louis), of which only 14 counties are designated as urban, while the other 101 are considered rural. Overall, the rate of mammography screening within the last two years for women over the age of 40 in Missouri is 71%, compared to a national average of 72% [23]. However, previous data has shown that within St. Louis City, the screening rate is 69.5% [10]. Additionally, studies have demonstrated that rates of breast cancer screening among rural women tend to be lower than those for urban women [24]. As a result, any efforts to increase mammographic screening rates in Missouri will face the challenge of confronting barriers encountered by women residing in both urban and rural populations. The mobile unit, affiliated with Siteman Cancer Center Breast Health Center, attempted to do just this by visiting sites in the St. Louis area (urban) as well as the rural Bootheel region of Missouri and addressing the Breast Cancer Community Partnership priorities.

Mobile mammography is a tool that has been used to reach underserved women in a diverse number of settings, including urban, rural, and mountain regions [25,26]. Community-academic partnerships between providers and community-based and faith-based organizations are an evidence-based approach for using mobile mammography to increase screening rates among racial and ethnic minorities, uninsured, medically underserved and women of low socioeconomic status [25]. Partnerships are effective in linking patients with mammography service as self-referral for mobile mammography may not optimally benefit or reach medically underserved women most in need of breast cancer screening [27]. Furthermore, results from a randomized controlled trial showed that women who were offered on-site mobile mammography in addition to health education were significantly more likely than those in the education-only group to undergo mammography sooner (within three months) [28].

Despite the use of mobile mammography vans as part of outreach efforts to increase breast screening rates in medically underserved communities, there is a lack of literature on the evaluation of these efforts and the development of evidence-based strategies. In this work, we demonstrate similarities and differences between rural and urban communities served by a single mobile mammography unit affiliated with the Siteman Cancer Center.

The Breast Health Center is housed within the Siteman Cancer Center and provides on-site service as well as off-site care through the use of a single mobile mammography van. Each day, this van goes to one location and serves women for the entire day. The van endeavors to visit three types of site locations in equal proportions throughout the month on a regular schedule. The site location categories are (1) community sites (grocery stores, YMCAs, and libraries), (2) corporate sites, and (3) outreach sites (FQHCs, well established health fairs, churches, etc.). Identification of community and outreach sites are determined based on data that identified “hot spots,” areas within St. Louis and surrounding counties that had particularly high rates of late-stage breast cancer at diagnosis. The van’s site schedule is continually evaluated to determine its effectiveness in breast cancer screening. The clinical service and PECaD and its analysts collaborate in evaluation efforts to optimally serve the community. In addition, community engagement has enhanced and informed the mammography van approach. For example, after disparities in late-stage breast cancer diagnosis and mortality were identified in the north county [29], following BCaP recommendations, we subsequently increased north county mammography van visits. Valeda’s Hope, which was created by a CRFT alumnus, and The Breakfast Club are two local community organizations we are partnered with which have worked to increase the number of women making use of the van in the region. Many strategies, including providing community education and scheduling opportunities ahead of the van’s visit, developing marketing materials that specifically invited uninsured women to be screened, and using breast health navigators to connect uninsured women with various breast cancer screening/treatment grant programs, have been employed to ensure the success of the van.

An outreach registry of patients served by the Breast Health Center (BHC) was created for program evaluation and planning purposes. Patients included in the database were uninsured or underinsured women that received screening care with funding from a Show Me Healthy Women grant, a program under the National Breast and Cervical Cancer Early Detection Program that provides free breast and cervical cancer screenings for women in Missouri with incomes at or below 200% of the federal poverty level who are aged 35–64 or older if they do not have Medicare Part B and have no insurance coverage for program services, or the Susan G. Komen Foundation grant. The registry data includes data from medical records and responses to a brief questionnaire completed at each visit. We examined data for the first screening visit of women that were seen on the mobile mammography van between April 2006 and December 2011. Data was examined by point of care (urban/rural) to assess the efficacy of mobile mammography as an outreach strategy in each of these environments.

Data analysis was conducted using SAS 9.3 (SAS Institute Inc., Cary, NC, USA); statistical significance was assessed at a *p*-value <0.05. Univariate analyses were used to examine categorical demographic characteristics such as age, income, race/ethnicity, education, employment status, marital status, insurance status, and living environment proxy (urban/suburban—St. Louis City/County, rural—Bootheel/other MO) using frequencies and percentages. Age was also examined continuously and summarized by the mean and standard deviation (Table 2). Bivariate associations were examined using chi-square tests to compare demographics between urban/suburban and rural living environment proxy. ArcGIS was used to map the number of women who received their first screening mammogram on the mobile unit and the number of times the unit visited each zip code between 2006 and 2011.

There were a total of 14,208 imaging records from 10,218 individual females in the BHC database from April 2006 to December 2011. During this period, 9789 women were enrolled in the registry for their first screening visit and 85% (*n* = 8289) received their care on the mobile mammography van. Thirty-four percent who presented were non-Hispanic White, 55% were non-Hispanic Black, 3% were Hispanic, and 9% were categorized as other race/ethnicity. The majority of women were in the 46–55 age group (mean = 52 years, SD = 8.9). Most (87%) had an annual income less than $20,000, had 12 years of education (45.1%) or less (31.1%), were unemployed (68%), uninsured (70%), and not married (72%). It was the first screening for 77% of women (Table 3).

The GIS maps show a wide outreach area for a single mobile unit. The highest numbers of women served were seen in the zip codes closer to the Siteman Cancer Center and in the rural Bootheel region of Missouri. However, the reach of mobile mammography was seen in women who reside in zip codes closer to Springfield, MO (approximately 200 miles from St. Louis) and across the state border in Illinois zip codes (Figure 1). Furthermore, it is interesting to note the large number of women served in the Bootheel region, despite the few number of mobile unit visits to that region.

The mobile mammography van improved access to breast cancer screening among a largely vulnerable uninsured population at risk for poor health due to being medically underserved, reaching women in both urban and rural areas in Missouri. Additionally, it successfully engaged economically disadvantaged women in medically underserved areas including women without health insurance who had never had a mammogram, and those with a family history of breast cancer.

While a larger percentage of the rural sample reported an income of greater than $20,000 (21.0%) as compared to the urban sample (11.4%), the vast majority of women in both groups were poor with incomes less than $20,000 (urban: 88.6% vs. rural: 79%). This is not an unexpected finding. However, it highlights an opportunity for both the education and recruitment of this particular population, as many of these women may be eligible for programs specifically for breast cancer screening based on income alone. For example, Show Me Healthy Women covers low-income women who are uninsured or underinsured. In addition, Medicaid in Missouri covers low-income individuals and families.

In this study, the majority of women in both groups had a high school education or less (urban: 75% vs. rural: 82.3%). This is actually a notable finding, because it means that the van is reaching a population of women that has traditionally been extremely underserved. Rates of mammographic screening in women aged 50–74 with less than a high school education was 69.4% and was 73.2% among those who have graduated from high school, both of which are beneath the national average [30]. Furthermore, prior research has also demonstrated that a decreased general understanding of breast cancer, and not just screening alone, contributes to lower mammography rates [31]. While our data demonstrates that this mobile mammography van is reaching a particularly vulnerable population, one must be mindful that this success is not from merely the presence of the van, but also likely from the significant community education that occurred as well.

This study has demonstrated that while there are certainly a few differences between urban and rural women served by the mobile mammography unit, the mobile unit predominantly serves a portion of the female population that is usually defined as medically underserved and “hard to reach”. These women are mostly poor, uninsured, and with limited educational backgrounds regardless of their zip code of service.

Support for components of the program have fluctuated over time, including changes with Medicaid, support for navigation, capacity, and organization priorities. To sustain decreases in the cancer burden, ongoing monitoring of services and resources is needed to ensure that access is sustained. Show Me Healthy Women faces barriers for providers to implement and maintain the program and is chronically underfunded, adding challenges to sustaining hospital engagement.

## 4. Discussion

To address disparities, community-based participatory strategies are an important tool to help identify and address health concerns while building trust in the community. Through our work with breast health community partnerships and a mobile mammography van, we successfully engaged with the community to help improve breast cancer outcomes.

In parallel, we have worked with our media and library systems to bring more information on breast health, the screening process, and the value of screening and follow-up of positive screening results, etc. to the community [32]. The media includes local and regional papers.

Continuous evaluation of efforts is crucial. For example, early in our work we did not evaluate community-based participatory principles resulting in us overlooking these and related partner concerns [33]. While we later developed a survey to assess adherence of PECaD to community-based participatory research and community engagement principles, establishing standards for evaluation would be useful to achieve project goals [34]. Evaluation also has helped us adapt to changing circumstances and identify gaps in our research. Reviewing systematic evaluation findings with the community resulted in expansion of membership to include more diverse voices [33].

For future directions, the goal is to use more complete data in mammograms to improve risk prediction and stratification independent of race/ethnicity. Mammograms have much detail in terms of texture, and these features may be precursors to breast lesions. Principal component measures can be used to capture texture features, density, and other measures consistent with images as intermediate markers of breast cancer [35]. Additionally, expanding breast navigation to extend through treatment and into survivorship could improve treatment completion and directly impact breast mortality rates. Increasing the range and extent of mobile units, such as this one, could be potentially effective in increasing the screening rates of not only women in Missouri, but also in other regions of this country and for other diseases. Expansion of our regional navigator network could similarly lessen the burden of breast cancer.

To reduce disparities and achieve health equity we have to work at multiple levels, sustain changes that we make to improve access to screening, diagnostic, and treatment services, bring bench science colleagues to embrace the challenges of disparities, and recognize and reward equity-promoting research and community-engaged research and service. Using the approach of collective impact, where different sectors work together with a common agenda and dedicated staff to solve social issues, we can help achieve these goals [36]. Only by doing so can we successfully reduce breast cancer burden in the populations most affected.

## 5. Conclusions

Through our community partnerships, we were able to help address disparities in breast cancer outcomes by expediting breast cancer screening and diagnosis and implementing patient navigation services. These efforts have resulted in a decrease in stage of diagnosis, improved access to mammography, and a sustained navigator network. Use of a mobile mammography unit also improved access to breast cancer screening among a largely vulnerable uninsured population, reaching women in both urban and rural areas in Missouri. This study demonstrates the potential for community-based participatory strategies to improve community health outcomes.

## Figures and Tables

**Figure 1 cancers-14-02550-f001:**
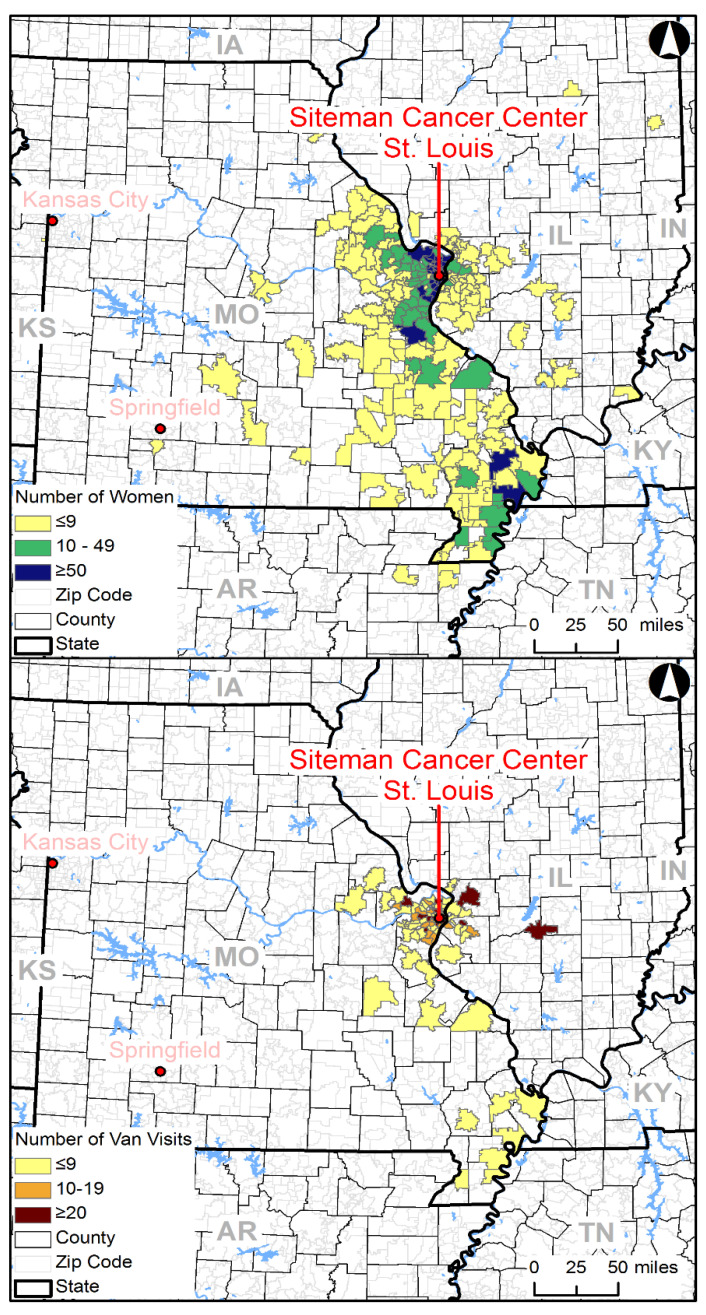
Number of women served (top) and van visits (bottom) by zip code.

**Table 1 cancers-14-02550-t001:** Breast cancer community partnership priorities.

Greater awareness (prevention and access to care)
2. Impact of culture
3. Build relationship/trust with community
4. Strategy to help patients keep appointments
5. Adherence to routing screening
6. Health literacy/communication
7. Evaluation

**Table 2 cancers-14-02550-t002:** Demographic characteristics of sample.

Total *n* = 8292	*n*	%
Age (mean, SD)	(52.21, 8.86)	
21–39	82	1.0
40–45	2056	24.8
46–55	3517	42.4
56–65	2016	24.3
66+	621	7.5
Annual Income 3-levels (*n* = 604)		
<$10,000	283	46.9
$10,000–$20,000	240	39.7
$20,000+	81	13.4
Race/Ethnicity (*n* = 8290)		
Non-Hispanic White	2785	33.6
Non-Hispanic Black	4529	54.6
Hispanic	267	3.2
Other	709	8.6
Highest Grade Completed ^1^ (*n* = 639)		
0–11	199	31.1
12	288	45.1
13–16	152	23.8
Currently Unemployed	5667	68.3
Not Married (*n* = 7694)	5558	72.2
No Insurance coverage (*n* = 8279)	5769	69.7
Urban proxy (*n* = 8176)		
St. Louis City/County—Urban/Suburban	6903	84.4
Bootheel/other MO—Rural	1273	15.6
Single visit	6367	76.8
Multiple visits	1925	23.2

^1^ Grades 0–11 represent a less than high school education, 12 represents graduating high school, and 13–16 represents attending some college or having a college degree.

**Table 3 cancers-14-02550-t003:** Bivariate association between demographics and urban versus rural comparison.

	STL City/County —Urban/Suburban		Bootheel/Other MO —Rural		Chi-Square *p*-Value
	*n*	%	*n*	%	
Age	6903		1273		38.37 df 4 *p* < 0.001
21–39	56	0.8	25	2.0	
40–45	1688	24.5	339	26.6	
46–55	2927	42.4	543	42.6	
56–65	1676	24.3	313	24.6	
66+	556	8.0	53	4.2	
Annual Income 3-levels	472		124		9.86 df 2 *p* = 0.007
<$10,000	232	49.2	46	37.1	
$10,000–$20,000	186	39.4	52	41.9	
$20,000+	54	11.4	26	21.0	
Highest Grade	512		124		3.69 df 2 *p* = 0.158
0–11	161	31.4	38	30.7	
12	223	43.6	64	51.6	
13–16	128	25.0	22	17.7	
Race 4-categories	6901		1273		1479.03 df 3 *p* < 0.001
Non-Hispanic White	1720	24.9	993	78.0	
Non-Hispanic Black	4376	63.4	122	9.6	
Hispanic	220	3.2	40	3.1	
Other	585	8.5	118	9.3	
Currently Unemployed	4701	68.1	899	70.6	3.16 df 1 *p* = 0.075
Married	1572	24.2	512	47.1	245.60 df 1 *p* < 0.001
No Insurance coverage	4644	67.3	1044	82.9	122.45 df 1 *p* < 0.001
Single visit	5190	75.2	1077	84.6	53.27 df 1 *p* < 0.001
Multiple visits	1713	24.8	196	15.4	

## Data Availability

Data are available upon reasonable request by contacting the corresponding author. All data relevant to the study are included in the article.

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
