# Peer review of "Strategies to Achieve Breast Health Equity in the St. Louis Region and Beyond over 15+ Years"

_cancers, 2022, doi:10.3390/cancers14102550_

Round 1

Reviewer 1 Report

The authors are to be commended for the significant work in developing and implementing community-engaged interventions to address disparities in breast health equity over the past 15 years. Given the persistent challenges in this area across the country, this topic is likely to be of high significance to the readers of this journal and would benefit the field of cancer prevention and control. With that said, there are a number of weaknesses with the current manuscript that make it difficult to realize the full potential of scientific contribution. First, the paper is organized in a way that is disjointed and describes multiple programs/activities (some in much more detail than others) that are not described as being particularly connected to one another. Also, this description does not logically follow chronologically from 15 years ago to today, including gaps in a discussion of work between 2003 and 2007 with the PECaD. If the programs described were/are part of a comprehensive approach, predicated on 7 priorities set out by the partnership, a brief description and/or figure would assist the reader in seeing how these strategies connect. A second challenge relates to the suggested premise of mammography being a major proven strategy in decreased breast cancer mortality (and presumably disparities) as pointed out on page 2: line 46, however, the authors note that Black women have much higher rates of mammography and higher mortality. Based on this contradiction, it is not clear how the strategies described here are/will contribute to breast health equity. Third, the title of the manuscript speaks to St Louis; however, the programs and data come from a much broader area, including parts of Illinois. Finally, the abstract suggests that the strategies used have resulted in "decreased stage of diagnosis and improved access to mammography;" however, results presented do not speak directly to either of these outcomes. What was the baseline? How was it measured? What was the actual (quantifiable) difference?

Author Response

Reviewer 1

The authors are to be commended for the significant work in developing and implementing community-engaged interventions to address disparities in breast health equity over the past 15 years. Given the persistent challenges in this area across the country, this topic is likely to be of high significance to the readers of this journal and would benefit the field of cancer prevention and control. With that said, there are a number of weaknesses with the current manuscript that make it difficult to realize the full potential of scientific contribution.

First, the paper is organized in a way that is disjointed and describes multiple programs/activities (some in much more detail than others) that are not described as being particularly connected to one another. Also, this description does not logically follow chronologically from 15 years ago to today, including gaps in a discussion of work between 2003 and 2007 with the PECaD. If the programs described were/are part of a comprehensive approach, predicated on 7 priorities set out by the partnership, a brief description and/or figure would assist the reader in seeing how these strategies connect.

Thank you for your comment. Per reviewer suggestion, we reordered the paragraphs in the breast health community partnership section to be in chronological order. We also added more text to show that PECaD was setting up its infrastructure between 2003-2007 and the Breast Cancer Community Partnership priorities were based off of work spanning several years including the period from 2003-2007. We added Table 1 to demonstrate how the different programs in the paper are predicated on these 7 priorities and added some text to clarify these connections. Additionally, we discussed some of the programs in more detail.

A second challenge relates to the suggested premise of mammography being a major proven strategy in decreased breast cancer mortality (and presumably disparities) as pointed out on page2: line 46, however, the authors note that Black women have much higher rates of mammography and higher mortality. Based on this contradiction, it is not clear how the strategies described here are/will contribute to breast health equity.

There are numerous factors that could explain the higher rates of mortality among Black women including disparities in healthcare access and quality, lack of quality insurance, higher late-stage diagnoses, and higher incidence of triple negative breast cancer, an aggressive subtype that is more prone to recurrence and less responsive to treatment. We added text to make this more clear. Although Black women have higher rates of mammography in the region, mammography is a proven method to reduce mortality across all races. In addition, we detail other strategies used to address disparities, such as using patient navigators to reduce barriers related to receiving all aspects of care, expediting breast cancer referrals, and offering community education.

“Factors that could explain these health inequities include disparities in healthcare access and quality, lack of quality insurance, higher late-stage diagnoses, and higher prevalence of triple negative breast cancer.”

Third, the title of the manuscript speaks to St Louis; however, the programs and data come from a much broader area, including parts of Illinois.

Thank you for pointing this out. We have revised our title to “Strategies to Achieve Breast Health Equity in the St. Louis Region and Beyond over 15+ Years” to reflect the expanded geographic coverage of our efforts.

Finally, the abstract suggests that the strategies used have resulted in "decreased stage of diagnosis and improved access to mammography;" however, results presented do not speak directly to either of these outcomes. What was the baseline? How was it measured? What was the actual (quantifiable) difference?

Per reviewer suggestion, to demonstrate the strategies used have resulted in improved access to mammography, we have added data on the study at the FQHC that shows the number of women receiving a mammogram before and after mammography and navigation services were implemented. 

“In addition, after one and two years of implementing mammography and navigation services, 17.7% and 27.6% of women over 40 years old received a mammogram, respectively, an increase from 12% in the nine-month period before the study started…In the following six months after the study was conducted, 18.7% of women aged 40 or older at the FQHC received a mammogram.”

Reviewer 2 Report

This is a generally very well written summary of the 15 year progress in St Louis to engage underserved communities in the region to breast health initiatives including screening mammography. The teams have made good progress which is discussed and they highlight multiple previous publications of their primary data in the references.

Major points:

  1. In the section on FQHC lines (179-189). The authors describe impressive increase in the number of women receiving a mammogram. However, it is not clear what they mean by 'increased mammographic capacity'. Were addition machines purchase, mammographers employed etc as a result of the increase in women coming forward? If so it would be useful to know how this was funded for an un/under-insured population.  I would also like to see evidence for the statement 'Importantly, these navigation and mammography services have been sustained'. I suspect there may have bene decline in reattendance for subsequent mammography and the figures for mammography, perhaps 1 and 3 years after the initial navigation would be very helpful in understanding the lasting success of the programme. A description of how the women were supported after initial navigation would also be helpful - is navigation an ongoing process or a one off interaction?
  2. The paragraph (lines 190-196) on late stage diagnoses suggests a non like for like comparison with all breast cancers included in the 2000 figures but only incident cancers in 2019. The success of a screening programme should be measured by ITT analysis and not just in those attending for screening. The incidence of stage 3/4 BC in Black women overall in 2019 should be included.
  3. Table 2 provides the summary of the important statistics and repeating them in the text (lines 289-308) seems superfluous to requirements.

Minor points:

sense check the sentence (lines78-82) "Ignited by alarmingly high breast cancer mortality rates in the St. Louis region, particularly in north St. Louis City and County among Black women, PECaD’s community advisory board advocated for research demonstrating the geographic region also showed high rates of late-stage breast cancer diagnosis which could help explain the mortality disparity." perhaps replace 'demonstrating the geographic region also' with 'which'.

Line 262/3 - please define 'select women'

Table 1. a definition of 'grades' of education would be useful for the non-American reader. It is also difficult to discern what these boundaries mean. I assume 0-11 means they did not attend complete high school but does 12 mean graduated from high school or did they just have to attend the year? redefining would help to compare wit the statement in line 337 'less than a high school education' 

Table 1 label 'unemployed' should probably be 'currently unemployed'  

The statement that ' the unit cannot operate in Illinois' (line 216) seems contrary to the GIS maps in which several counties of Illinoi were covered. Please  remove the statement or clarify

Bootheel region should be labelled on the GIS maps

line 321 please clarify the term 'vulnerable' 

Author Response

This is a generally very well written summary of the 15 year progress in St Louis to engage underserved communities in the region to breast health initiatives including screening mammography. The teams have made good progress which is discussed and they highlight multiple previous publications of their primary data in the references.

Major points:

  1. In the section on FQHC lines (179-189). The authors describe impressive increase in the number of women receiving a mammogram. However, it is not clear what they mean by 'increased mammographic capacity'. Were addition machines purchase, mammographers employed etc as a result of the increase in women coming forward? If so it would be useful to know how this was funded for an un/under-insured population. I would also like to see evidence for the statement 'Importantly, these navigation and mammography services have been sustained'. I suspect there may have bene decline in reattendance for subsequent mammography and the figures for mammography, perhaps 1 and 3 years after the initial navigation would be very helpful in understanding the lasting success of the programme. A description of how the women were supported after initial navigation would also be helpful - is navigation an ongoing process or one off interaction?

Thank you for pointing these issues out. We reworded ‘increased mammographic capacity’ to say ‘promoted mammogram utilization’ and clarified that a patient navigator, mammography technologist, and data coordination assistant were hired through American Recovery and Reinvestment Act funding. We added more information to support that these navigation and mammography services have been sustained as an ongoing process including follow-up visits. Although we do not have data from 1 and 3 years after the initial navigation, we added six-month post-study data.

“Given 88% of eligible women at the FQHC had not received a mammogram in 2008, we received American Recovery and Reinvestment Act funding to implement and evaluate breast navigation among women due or overdue for mammography from 2009-2011. Using these funds allocated for creating jobs and sustainability, we hired a patient navigator, mammography technologist, and data coordination assistant at this underserved location. Patient navigators connected women with mammography, diagnostic, and treatment services, guided women through all aspects of care including follow-up visits and contacted women who had lost contact with the clinic.  Patient navigation promoted mammogram utilization with an increased number of mammograms provided overall…In addition, after one and two years of implementing mammography and navigation services, 17.7% and 27.6% of women over 40 years old received a mammogram, respectively, an increase from 12% in the nine-month period before the study started. Navigation services continue to be offered alongside clinical care as an ongoing process with the created roles now funded by the FQHC. In the following six months after the study was conducted, 18.7% of women aged 40 or older at the FQHC received a mammogram.”

  1. The paragraph (lines 190-196) on late stage diagnoses suggests a non like for like comparison with all breast cancers included in the 2000 figures but only incident cancers in 2019. The success of a screening programme should be measured by ITT analysis and not just in those attending for screening. The incidence of stage 3/4 BC in Black women overall in 2019 should be included.

            We have expanded text but not added much more given that the numbers of new cases confirmed by tumor registry vary by year, and proportion by stage fluctuates more when we break out race subgroups. Our approaches to analysis have included smoothed curves and absolute annual proportions both showing the same trend of decreasing late stage but consistent gap between Black and White women. We clarified the sentence in the abstract.

  1. Table 2 provides the summary of the important statistics and repeating them in the text (lines 289-308) seems superfluous to requirements.

Per reviewer suggestion, we have deleted the paragraph summarizing the results of Table 2.

Minor points:

sense check the sentence (lines78-82) "Ignited by alarmingly high breast cancer mortality rates in the St. Louis region, particularly in north St. Louis City and County among Black women, PECaD’s community advisory board advocated for research demonstrating the geographic region also showed high rates of late-stage breast cancer diagnosis which could help explain the mortality disparity." perhaps replace 'demonstrating the geographic region also' with 'which'.

Per reviewer suggestion, we have replaced ‘demonstrating the geographic region also’ with ‘which’.

“Ignited by alarmingly high breast cancer mortality rates in the St. Louis region, particularly in north St. Louis City and County among Black women, PECaD’s community advisory board advocated for research which showed high rates of late-stage breast cancer diagnosis which could help explain the mortality disparity.”

Line 262/3 - please define 'select women'

We have removed the word ‘select’ and instead described the eligibility criteria for Show Me Healthy Women.

“Patients included in the database were uninsured or underinsured women that received screening care with funding from a Show Me Healthy Women grant, a program under the National Breast and Cervical Cancer Early Detection Program that provides free breast and cervical cancer screenings for women in Missouri with incomes at or below 200% of the federal poverty level who are aged 35 – 64 or older if they do not have Medicare Part B and have no insurance coverage for program services, or the Susan G. Komen Foundation grant.”

Table 1. a definition of 'grades' of education would be useful for the non-American reader. It is also difficult to discern what these boundaries mean. I assume 0-11 means they did not attend complete high school but does 12 mean graduated from high school or did they just have to attend the year? redefining would help to compare wit the statement in line337 'less than a high school education'

Thank you for bringing this to our attention. We have added a footnote explaining these boundaries.

“Grades 0-11 represent a less than high school education, 12 represents graduating high school, and 13-16 represent attending some college or having a college degree”

Table 1 label 'unemployed' should probably be 'currently unemployed' 

We have changed ‘unemployed’ in the tables to say ‘currently unemployed’ instead.

The statement that ' the unit cannot operate in Illinois' (line 216) seems contrary to the GIS maps in which several counties of Illinoi were covered. Please remove the statement or clarify

Thank you for bringing this to our attention. The statement has been removed.

line 321 please clarify the term 'vulnerable'

We have clarified that the women are vulnerable due to being medically underserved.

“The mobile mammography van improved access to breast cancer screening among a largely vulnerable uninsured population at risk for poor health due to being medically underserved, reaching women in both urban and rural areas in Missouri.”

Round 2

Reviewer 1 Report

The authors have thoughtfully and sufficiently responded to all prior critiques.